# Developmental Genetic Basis of *Hoxd9* Homeobox Domain Deletion in *Pampus argenteus* Pelvic Fin Deficiency

**DOI:** 10.3390/ijms241411769

**Published:** 2023-07-21

**Authors:** Shun Zhang, Xiaodong Zhang, Cheng Zhang, Shanliang Xu, Danli Wang, Chunyang Guo

**Affiliations:** 1School of Marine Science, Ningbo University, Ningbo 315211, China; 18186470622@163.com (S.Z.); xd030495@163.com (X.Z.); chengzhangcrab@163.com (C.Z.); xushanliang@nbu.edu.cn (S.X.); 2National Engineering Research Laboratory of Marine Biotechnology and Engineering, Ningbo University, Ningbo 315211, China; 3Key Laboratory of Applied Marine Biotechnology, Ningbo University, Chinese Ministry of Education, Ningbo 315211, China; 4Key Laboratory of Green Mariculture (Co-Construction by Ministry and Province), Ministry of Agriculture and Rural, Ningbo University, Ningbo 315211, China

**Keywords:** limb positioning, *Hoxd9* gene, domain deletion, *Pampus argenteus*, evolution, missing pelvic fins

## Abstract

*Pampus argenteus* is important for commercial fishery catch species and is an emerging target for aquaculture production. Notably, *P. argenteus* has a bizarre morphology and lacks pelvic fins. However, the reason for the lack of pelvic fins remains unclear, ultimately leading to frequent upside-down floating of *P. argenteus* during breeding and marked consumption of physical energy. Some lineages, including whales, fugu, snakes, and seahorse, independently lost the pelvic appendages over evolutionary time. Do different taxa employ the same molecular genetic pathways when they independently evolve similar developmental morphologies? Through analysis of the gene responsible for appendage localization, Hoxd9, it was discovered that the Hox domain was absent in the *Hoxd9* gene of *P. argenteus*, and the *Hoxd9b* gene lacked the Hox9 activation region, a feature not observed in the *Hoxd9* gene of other fish species. Interestingly, those distinctive characteristics are not observed in the *Hoxd9* gene of other fish species. To determine the association between the *Hoxd9* gene characteristics and the pelvic fin deletion in *P. argenteus*, the full-length cDNA of the *Hoxd9a* gene was cloned, and morphological observations of the species’ juveniles were performed using stereomicroscopy and scanning electron microscopy. Thereafter, the tissue localization of Hoxd9a in the species was analyzed at the gene and protein levels. Based on the results, deletion of the Hoxd9a structural domain possibly leads to disruptions in the protein translation and the pelvic fin localization in *P. argenteus* during its early ontogenetic developmental stage, resulting in the absence of pelvic fins.

## 1. Introduction

The development of vertebrate appendages has been reported to have gone through three main stages: limb positioning, limb bud initiation, and limb bud growth [1]. A zone of polarization activity (ZPA) exists in the fins and limbs and is composed of a group of specialized mesenchymal cells located at the posterior edge of the bud that can control the anterior and posterior limb differentiation via sonic hedgehog (Shh) protein. The apical ectodermal ridge (AER) is a temporary structure in the fin of teleost fish. Soon after this structure appears, it elongates to form an apical ectodermal fold (AEF), in which the dermal fin is differentiated by rays. Zeller described the functions of the signal centers (ZPA and AER) from the front to the rear (AP) and from the proximal to the distal axes (PD) [2]. The expression pattern analysis of the appendage localization-related genes discovered that the lateral plate mesoderm (LPM) cells exhibit continuous proliferation, accompanied by a gradual accumulation of inducing factors that eventually develop into lateral limb buds. The ectodermal cells react with the signal cascade, resulting in limb bud sprouting along the edges of the front and back axes of the embryo and horizontal development in the axial direction to form the AER [3]. By fibroblast growth factors (Fgfs) into mesenchymal cells, limb buds can grow further [4]. The positioning of the forelimb, interlimb region, and hindlimb in tetrapods is known to be regulated by the expression *Hoxb9*, *Hoxc9*, and *Hoxd9* genes in the lateral plate mesoderm, and the positioning of the forelimb is associated with the anterior expression border of *Hoxc6* in the paraxial mesoderm [1]. The molecular mechanism regulating appendage development is a highly conserved genetic network that mainly includes limb localization genes (*Hoxc6*, *Hoxd9*), limb bud inducing genes (*Pitx1*, *Tbx4*, *Tbx5*), and limb bud growth-related genes (*Wnt*, *Shh*, *Fgf10*).

The molecular genetic basis of limb developmental disorders has been examined in few vertebrate species. Cetaceans, despite lacking pelvic fins, have the ability to initiate the formation of pelvic fin buds and express Fgf8 to establish the apical ectodermal ridge (AER). However, the gene signal responsible for this process is not sustained, and hind limb buds experience the absence of Shh, a crucial factor in establishing the zone of polarizing activity (ZPA). As a consequence, pelvic fin degeneration occurs [5]. The in situ hybridization analysis of *Takifugu rubripes* demonstrated the absence of *Hoxd9a* expression in the presumed pelvic fin region, resulting in the inability to activate the signaling pathway responsible for fin initiation and growth [1]. In contrast, the *Tbx4* gene was not found following whole-genome sequencing of the seahorse. The *Tbx4* gene is a determinant of pelvic fin initiation and, therefore, directly contributes to the loss of pelvic fin structure in the seahorse (Syngnathidae) [6]. In snakes, the tissue expression activity of the Hoxc6/Hoxc8 genes expands posteriorly along the body axis, encompassing the most anterior somites and extending to the level of the cloaca and hindlimbs. Likely, it triggers a shift in the entire snake’s trunk towards thoracic/lateral characteristics, thereby preventing the axial positioning of limb buds and their subsequent development [7]. It is hypothesized that the loss of appendages can be attributed to the abnormal expression or deletion of crucial genes involved in limb development, although the specific genes responsible for this phenomenon may vary among species. 

Hox genes were first discovered by Lewis (1978) in a study on homozygous mutations in *Drosophila* [8]. A total of 39 Hox genes have been identified and organized into four separate clusters according to their chromosomal localization, i.e., chromosome pair number 7 (HoxA), 17 (HoxB), 12 (HoxC), and 2 (HoxD). *Hox* genes, regardless of fragment size, have a conserved sequence consisting of 180 bases and encode a polypeptide region of 60 amino acids called the homologous structural domain (HD). HD-containing proteins are transcriptional regulators that can activate or repress the expression of target genes [9,10,11]. HD-containing genes play a role in encoding DNA-binding proteins, regulating gene expression, and controlling morphogenesis and cell differentiation [12]. Available studies indicate that the nested HD genes within the Hox family play a fundamental role in vertebrate embryonic morphogenesis by instructing cells about their positional identity along the main body axis, ensuring the appropriate formation of body parts [13].

Based on earlier studies, the most anterior gene, *Hoxd9*, is expressed in the lateral plate mesoderm to the thorax prior to limb sprouting [14]. During the initiation of limb sprouting, the expression of *Hoxd9* is sustained, while the *Hoxd10-Hoxd13* genes are activated in a sequential manner, resulting in a nested pattern of spatially and temporally defined expression domains along the anterior-posterior axis of the fin and limb. Among these genes, *Hoxd9a* occupies the largest region, followed by *Hoxd10a*, *Hoxd11a*, *Hoxd12a*, and *Hoxd13a* [15]. Extensive evidence has underscored the functional significance of the HoxA and HoxD gene clusters within the Hox gene family, highlighting the critical role of mutations in the acquisition or loss of their functions in mammalian organisms. It was demonstrated that the complete loss of either HoxA or HoxD transcriptional activity might cause profound limb congenital malformations during the ontogenetic development of the mouse, which results in complete retardation of limb formation [16]. However, the absence of the HoxB and HoxC clusters does not lead to limb abnormalities [17]. Therefore, HoxA and HoxD are hypothesized to play an indispensable role in limb development in vertebrates [18]. Further studies have indicated that the deletion of a specific region within the Hox cluster eliminates the shared enhancer, leading to erroneous expression patterns of the remaining Hox genes [19]. Therefore, the destruction of a single Hox gene or a combination of Hox genes can selectively disrupt their respective growth and development processes. For example, a defect in the Hoxb5 gene function causes incorrect placement of the forelimb along the anteroposterior axis of the body [20]. In turn, mutations in the *Hox8* gene result in tail-shifted hind limbs [21]. In some cases, duplication, translocation, inversion, and deletion of the *hoxd* gene also cause mesodermal dysplasia of the upper and lower limbs [22]. In mice, mutations in the homologous *Hox9* and *Hox10* genes result in short sarcomeres (including the humerus and femur) [23,24]. Other studies demonstrated that mutations in *Hoxd11*, *Hox12*, *Hoxb13*, and *Hoxd13* are the main factors responsible for alterations in the number of phalanges and the length of fingers [24,25,26]. Tanaka’s study [1] demonstrated that the *Hoxd9* gene plays a crucial role in the ontogenetic development of pelvic fin positioning in sticklebacks (Gasterosteidae), and inhibition of its transcriptional activity led to complete disruption of pelvic fin formation in the *Takifugu rubripes*. Hence, undertaking a study to explore the potential involvement of the *Hoxd9* gene in the loss of pelvic fins in *Pampus argenteus* seems justified, as it aligns with the objective of gaining a deeper understanding of the evolutionary mechanisms related to limb loss in vertebrates.

*Pampus argenteus* (*P. argenteus*) belongs to Perciformes, Stromateidae, and is one of the most economically important marine fish in China owing to its delicate meat, low intermuscular stings, high nutritional quality, as well as high market value and demand. During the 1980s, a series of domestic research efforts on *P. argenteus* biology was conducted, resulting in significant advancements in areas such as artificial breeding, artificial seedling culture, ecology, histology, and species’ pathology [27,28,29]. *P. argenteus* is oval in shape and flat on its side, with one dorsal fin and short fin spines. The anal and dorsal fins of the species have the same shapes. The pectoral fin is long, while the caudal fin is bifurcated; however, there is no pelvic fin, which contributes to the species’ reduced balancing abilities in the water. *P. argenteus* frequently exhibits a behavior of rubbing against the walls of the pond, resulting in body scratches. Based on classical ichthyology theories, “the pelvic fin possibly supports dorsal and anal fins to maintain the balance of the body, and reinforces of fish’s abilities to lift and turn”. Therefore, the physical energy consumption of *P. argenteus* is large, increasing the difficulty associated with the artificial breeding of *P. argenteus*. In the present study, we successfully cloned the full-length cDNA of the *Hoxd9* gene in *P. argenteus* and employed scanning electron microscopy (SEM) to investigate the presence of pelvic fin buds in *P. argenteus* larvae. In addition, fluorescence in situ hybridization and immunofluorescence techniques were employed to locate the distribution of the Hoxd9 mRNA and protein expression in *P. argenteus* to provide a theoretical basis for the study of *P. argenteus* pelvic fin loss.

## 2. Results

### 2.1. Cloning and Sequence Analysis of the Full-Length cDNA of the Hoxd9 Gene from P. argenteus

Through the cloning of the *Hoxd9a* gene in *P. argenteus*, it was discovered that the *Hoxd9a* gene in *P. argenteus* is 180 bp shorter than its counterparts in other fish species. Smart and Signal P-5.0 analysis revealed the 180 bp deletion is the homeobox domain (Figure 1). The full-length nucleotide sequence and deduced amino acids of Hoxd9a of *P. argenteus* are shown in Figure 2. The length of *Hoxd9a* cDNA was 1602 bp (GenBank accession: MZ475051), and the 633 bp open reading frame (ORF) was found to encode 210 amino acids. Moreover, the protein molecular weight was 22.92 kDa, and the theoretical isoelectric point (pI) was 7.21. Based on PSORT analysis, the Hoxd9a protein was found to be mainly located in the endoplasmic reticulum, followed by the mitochondria and nucleus. The secondary structure was mainly composed of 3.33% alpha helix, 1.42% extended strand, and 95.25% random coil. The tertiary structure of the protein was obtained by analyzing the amino acids using SWISS-MODEL (Figure 3).

### 2.2. Amino Acid Sequence Alignment and Homology Analysis of Hoxd9 in P. argenteus

ClustalW was used to align the Hoxd9a/b of *P. argenteus* with that of the other six fish (Figure 4). *P. argenteus* Hoxd9a had the highest homology (67%) with *Sander lucioperca, Larimichthys crocea*, and *Oreochromis niloticus*, followed by *Sparus aurata*, *Salarias fasciatus* (66%), and *Takifugu rubripes* (65%).

The MEGA X (10.0.2) software was used to analyze the molecular phylogeny of the Hoxd9 proteins from 28 different species in the NCBI database. The evolutionary relationship between Hoxd9 and other species was examined via the construction of a phylogenetic tree (Figure 5). The phylogenetic tree was constructed using maximum parsimony, and the algorithm selected neighbor-joining.

Based on the phylogenetic tree, Hoxd9 is divided into two subgroups: Hoxd9a and Hoxd9b. *P. argenteus* Hoxd9a was found to be genetically closest to *Nothobranchius furzeri* (XM_015966458.1). The closest genetic distance between *P. argenteus* Hoxd9a and *Nothobranchius furzeri* (XM_015966458.1), *Salarias fasciatus* (XM_030111970.1), *Takifugu rubripes* (NC_042285.1), *Sparus aurata* (XM_030427561.1), *Larimichthys crocea* (NC_040028.1) was found to be in the same subgroup, while the closest genetic distance between *P. argenteus* Hoxd9b and *Fundulus heteroclitus* (XP_035982622), *Kryptolebias marmoratus* (XP_024864393), *Haplochromis burtoni* (XP_005922527), *Takifugu rubripes* (XP_003963854) was found to be in another subgroup.

### 2.3. Expression of Hoxd9a Gene in Different Tissues and Ontogenetic Developmental Periods in P. argenteus

Expression of *P. argenteus Hoxd9a* mRNA relative to the expression of the reference gene *18S rRNA* at different tissues and ontogenetic developmental stages was determined using qPCR (Figure 6). *Hoxd9a* mRNA expression began to appear in the early stages of embryonic development, showing an initial increase and reaching its peak level during the gastrula stage (*p* < 0.05). The relative expression then showed a “U” trend and was lowest at 1–13 days (*p* < 0.05, Figure 6a). The analysis of relative expression levels of *Hoxd9a* in various tissues of *P. argenteus* indicated the absence of *Hoxd9a* expression in the brain and eyes. The pectoral fin has the highest expression of *Hoxd9a* (*p* < 0.05), followed by muscle and the caudal fin. *Hoxd9a* was also found in the abdominal epithelium, albeit at a much lower level than in other tissues (*p* < 0.05, Figure 6b).

### 2.4. Morphological Observation of P. argenteus Larvae and Juveniles

Newly hatched larvae of *P. argenteus* had a full length of 3.26 ± 0.33 mm, with well-developed fin folds and dorsal pelvic fin folds of nearly twice the body width. The 1 day larvae had a length of 3.86 ± 0.25 mm, with a nearly round pectoral fin primordium (Figure 7a). The 4 days larvae had a length of 4.85 ± 0.28 mm, with fan-shaped pectoral fins and strong rowing. The length of 6 days larvae was 4.98 ± 0.1 mm, and the tail vertebra began to be slightly cocked. The total length of the 7–10 days larvae was 5.05 ± 0.23 mm, the cells accumulated under the upturn of the tail vertebra, and the abdomen near the heart appeared as a small bulge (Figure 7b,c, red dotted circle). The length of the 13 days larvae was 6.55 ± 0.15 mm. At this developmental stage, pectoral fin rays were observed, caudal fin basal fin bone primordium formed, the dorsal fin and anal fin began to accumulate cells, the abdomen near the heart was prominent, and local bulges caused by ossification of the hipbone were observed (Figure 7d,e). The 16 days juveniles had a total length of 7.14 ± 0.11 mm. Their body gradually became opaque, the caudal fin rays began to form, and the dorsal and anal fin bone primordia formed (Figure 7f,g). The 19 days juveniles had a total length of 8.59 ± 0.16 mm, with rapid development of each fin, 9 pectoral fin rays, 21 caudal fin rays, 28 dorsal fin rays, and 25 anal fin rays. Notably, the abdomen bulge could still be observed (Figure 7h,i). The 22 days juveniles had a total length of 9.43 ± 0.13 mm, their dorsal and anal fins were found to reach the thinnest part of the caudal stalk, and the fin membrane disappeared. The 33 days juveniles had a total length of 12.07 ± 0.31 mm, with slightly concave caudal fins, 21 caudal fin rays divided into 5 segments, and more than 40 dorsal fin rays. The 37 days juveniles had a total length of 13.37 ± 0.38 mm, 21 pectoral fins, and a deep concave caudal fin. Visual observation revealed that the caudal stalk was transparent. At this time, pelvic fins had not yet appeared. Through micro CT scanning, a comparison of the bone structure of *P. argenteus* revealed that the observed bulges corresponded to the ossification of the hipbone (Appendix A).

SEM analysis was carried out using the larvae and juveniles of *P. argenteus* at 1 day, 3 days, 7 days, 13 days, 16 days, 19 days, and 25 days (Figure 8). No signs of formation initiation or occurrence of pelvic fin buds were observed at any developmental stage (red dotted circle in the figure). An expanded view of the abdomen can be found in Appendix A.

### 2.5. Fluorescence In Situ Hybridization

The results of positive staining revealed the tissue-specific expression pattern of the Hoxd9a mRNA at different developmental stages of *P. argenteus*. The *Hoxd9a* mRNA was mainly expressed in the head (except the eyes) and trunk of 1 day larvae (Figure 9A). In 7 days old larvae, the *Hoxd9a* mRNA was mainly expressed in the foregills and tail (Figure 9B). In 13 days old larvae, *Hoxd9a* mRNA was mainly detected in the middle trunk and tail (Figure 9C). In 16 days juveniles, *Hoxd9a* mRNA was mainly distributed in the upper lateral line (Figure 9D). In the 19 days juveniles, *Hoxd9a* mRNA was mainly distributed in the gill and upper part of the trunk (Figure 9E), with no obvious fluorescence signal in the abdomen at different developmental stages (red dotted circle). There were no hybridization signals in the control fish group at all developmental stages (Appendix A).

### 2.6. Immunofluorescence

Based on the immunofluorescence results, Hoxd9a was mainly distributed in the head (except eyes) and trunk of 1 day larvae (Figure 10A). In 7 days larvae, Hoxd9a was mainly distributed in the gills, trunk, and back half of the abdomen (Figure 10B), while in 13 days larvae, Hoxd9a was mainly distributed in the middle torso and tail (Figure 10C). In the 16 days juveniles, Hoxd9a was mainly distributed in the gill, front trunk, and caudal fin (Figure 10D), whereas in 19 days juveniles, Hoxd9a was distributed in the head, gill, and posterior abdomen (Figure 10E), with no obvious fluorescence signal in the abdomen at different developmental stages (red dotted circle). Notably, there was no signal in the control group in the five developmental stages (Appendix A).

## 3. Discussion

The Hox gene not only plays a crucial role in determining the axial pattern during embryonic development but also contributes to the regionalization of the lateral mesoderm, specifying the proper positions of the forelimbs, the sizes of the areas between limbs, and the hind limbs [14]. In teleost fish, the pelvic fin bud develops later than the pectoral ones. For example, the pectoral fin bud of zebrafish begins to develop at 26 h after fertilization, while the pelvic fin bud develops during metamorphosis 3 weeks after fertilization [30]. *P. argenteus* has no pelvic fin; however, its pectoral fin primordium appeared at 30 h after hatching, and its rowing was strong at 86 h after hatching. The accumulation of caudal fin cells in *P. argenteus* occurred between 9 and 11 day after hatching, with the primordium of the caudal fin becoming visible 13 days after hatching. The dorsal fin and anal fin cells of *P. argenteus* accumulated 16 days after hatching, and their fins were fully developed 16–22 days after hatching. In this study, the larvae and juveniles of *P. argenteus* at 1 day, 7 days, 13 days, 16 days, and 19 days were subjected to morphological observation and SEM analysis. No pelvic fin bud appeared in the entire mesoderm of the lateral plate from the pectoral fin formation area to the tail at each developmental stage, which contributed to the species’ reduced balancing abilities in the water. *P. argenteus* frequently exhibits a behavior of rubbing against the walls of the pond, resulting in body scratches. The pelvic fin possibly supports dorsal and anal fins to maintain the balance of the body and reinforces the fish’s abilities to lift and turn. According to a previous study [31], the silencing of the *Pitx1* gene promoter during the early ontogenetic development stage of *Gasterosteus aculeatus* leads to the improper formation of the pelvic fin bud, resulting in complete retardation of pelvic fin development in the species. When combined with earlier research on the *P. argenteus Hoxc6*, *Tbx4/5*, *Pitx1*, *Fgf8/10*, and *Shh* genes, we discovered that these genes were not the key regulators of *P. argenteus* pelvic fin deletion. Studies in zebrafish have demonstrated that the knockdown of the *Tbx4* gene leads to pelvic fin deficiency, while in *P. argenteus*, *Tbx4* expression occurs in an undisturbed state during its ontogenetic development [6]. Intriguingly, our study revealed that the upstream gene of *Tbx4* in *P. argenteus*, *Hoxd9*, lacks the homeodomain. It is reasonable to infer that the pelvic fin bud is experiencing problems in the positioning stage, and thus, the development of the pelvic fin bud is not activated.

The *Hox* gene encodes a highly conserved transcription factor family that significantly affects many cell processes, including proliferation, apoptosis, cell shape, and cell migration. The *Hox* gene contains a conserved sequence of 183 bp, which encodes a nuclear protein called a homologous protein [32]. However, in the *P. argenteus* genome, the 183bp conserved sequence is absent from Hoxd9a, whereas Hoxd9b lacks the Hox9 activation region. The HOX protein contributes to anterior and posterior axis modeling during embryonic development [33], and the *Hox* gene can control normal cell proliferation and differentiation in different tissues [34]. There are four *Hox* gene clusters in the HOX family: *HoxA*, *HoxB*, *HoxC*, and *HoxD*; these clusters play key roles in the modeling process of vertebrate shaft bones and limbs by changing the protein sequences and expression patterns [35,36]. In vertebrates, the *HoxA* and *HoxD* cluster genes are mainly responsible for the formation of the forelimb pattern, while the *HoxC* cluster genes are mainly involved in the formation of hind limb patterns [19]. Other studies have also shown the crucial role of *HoxD9-13* genes in the limb formation of vertebrates [25,37]. *Hoxd9* is one of the *HoxD* genes located closest to the 3’ end of the chromosome, which is mainly involved in the formation process of forelimbs and central bones, and is very important for embryo segmentation and limb bud development [19]. 

The full-length cDNA sequence of the *Hoxd9a* gene in *P. argenteus*, containing a 633 bp ORF encoding 210 amino acids, was successfully obtained through homologous cloning and RACE technology. In the multi-sequence alignment of homologous sequences and SWISS-MODEL analysis of fish, it was observed that approximately 180 bp of the nucleotide sequence at the 3’ end of the ORF of the *Hoxd9a* gene in *P. argenteus* was absent. Comparing it with other sequences, this region was identified as the homeobox domain. Russell and Mario [38] clearly revealed that changes in *Hox* gene function are closely related to the loss of homeobox DNA sequences in the regulatory regions of downstream genes, leading to changes in the intensity, time, and spatial domain of *Hox* gene expression patterns. According to the literature [39,40], the deletion of the whole HoxD gene cluster or the 5-terminal of the cluster is related to severe limb and genital deformity. Moreover, mutation of the *Hox* gene usually leads to obvious loss of cell proliferation and limb hypoplasia. Zhu et al. [41] proposed that the *Hox* gene, which is closely related to embryonic development, is abnormally expressed in both dysplasia and malignant lesions, suggesting that abnormal expression of the *Hox* gene plays an important role in the loss of the pelvic fin of *P. argenteus*. Based on the literature [42,43], most HOX proteins have evolutionarily conserved Ser-Ser-Tyr-Phe (SSYF) motifs and homeobox domains necessary for functional activities. Therefore, deletion of the SSYF motifs and homeobox domains will lead to a decrease in ectopic activation at corresponding sites and a large number of ectopic expressions at other sites. Tour et al. [43] provided confirmation that the deletion of the N-terminal sequence of Ultrabithorax (Ubx), a HOX protein, leads to a significant reduction in its transcriptional activation function. Additionally, when the region of the Ubx variant containing the SSYF motif is deleted, it undergoes a transformation from an activator to a transcriptional inhibitor. In this study, the homeobox domain and SSYF motifs were simultaneously deleted from the *Hoxd9a* C-terminus of *P. argenteus*; such deletion was expected to reduce or block its transcriptional activation function, thereby inhibiting the occurrence of pelvic fin buds of *P. argenteus* to a certain extent.

A recent study has suggested that *Hoxb9* and *Hoxc9* genes may serve a compensatory role in limb positioning for tetrapods, as the absence of a pelvic phenotype was observed in Hoxd9 knockout mice [44]. This compensation would not be expected for forelimb patterning in tetrapods because, in contrast to the pelvic limb, Hoxd9 is expressed in the forelimb during the limb positioning phase in the absence of the other three Hox9 group genes [1]. The common phenotype of pelvic reduction can result from disruptions at different phases of the conserved genetic pathway for appendage development, including the outgrowth phase in pythons [7], the initiation phase in independently evolved stickleback populations [31,45], or the positioning phase in fugu [1].

mRNA localization is a common occurrence observed in oocytes, developing embryos, and differentiated somatic cells. The growing body of evidence suggests that mRNA localization plays a crucial role in intracellular protein targeting, facilitating local protein biogenesis [46,47]. However, protein localization has always been regarded as an evolutionary advantage in eukaryotes, which isolate intracellular membranes to achieve specific functions [48]. In most cases, mRNAs have a close spatial association with their protein products [49]. In numerous cells, mRNA is distributed in discrete locations within the cytoplasm, aligning with protein translation and facilitating precise spatial and temporal regulation of protein function. The subcellular localization of proteins is crucial for their proper functioning. Multiple studies have highlighted the evolutionary conservation of mRNA localization as a mechanism that regulates protein localization. In the case of Hoxd9, it plays a significant role in the localization of limb buds. *Hoxd9a* relative expression was lowest in 1–13 days larvae, which corresponded to the expected occurrence of *P. argenteus* pelvic fins at 1–13 days. However, the location of Hoxd9a mRNA and protein at different developmental stages is not the same in *P. argenteus*. In 1 day larvae, Hoxd9a mRNA and protein were basically found at the same location, mainly on the head (except the eyes) and trunk. In 7 days larvae, *Hoxd9a* mRNA was located in the front operculum and tail, while Hoxd9a protein was located in the gills, trunk, and the back half of the abdomen. In 13 days larvae, *Hoxd9a* mRNA was localized in the middle of the trunk and tail, while Hoxd9a protein was localized in the middle torso and tail. In 16 days juveniles, *Hoxd9a* mRNA was localized in the upper lateral line, while Hoxd9a protein was localized in the gill, front trunk, and caudal fin. In 19 days juveniles, *Hoxd9a* mRNA was localized in the gill and upper part of the trunk, while Hoxd9a protein was only localized in the gill of the head and back of the abdomen. Analysis of mRNA and protein localization revealed that, despite the presence of *Hoxd9a* mRNA and Hoxd9 protein in various developmental stages of *P. argenteus*, no discernible fluorescence signals were detected in the region associated with pelvic fin formation.

Unfortunately, laboratory manipulations to test these hypotheses are not feasible with *P. argenteus* embryos at present. Nonetheless, transgenesis experiments in zebrafish offer a viable option to investigate whether the absence of cis-acting regulatory elements responsible for flank and pelvic expression of Hoxd9, as found in zebrafish genes, might be the case in *P. argenteus*.

## 4. Materials and Methods

### 4.1. Experimental Materials

*P. argenteus* was obtained from the *P. argenteus* breeding pond of Xiangshan Harbor Hatchery Co., Ltd. (Ningbo, China). Ten healthy *P. argenteus* individuals were selected at different developmental stages, including 1 day (pectoral fin primordium appeared), 3 days (powerful pectoral fins), 7 days (caudal fin cells appeared), 13 days (caudal fin primordium appeared), 16 days (dorsal fin and anal fin primordium appeared), and 19 days (each fin was intact). Nine types of tissue samples were collected from fully developed mature *P. argenteus* (50–70 g): heart, brain, eye, muscle, pectoral fin, abdominal epithelium, dorsal fin, anal fin, and caudal fin. Samples for fluorescence in situ hybridization were fixed in 4% paraformaldehyde (PFA) solution for 12 h, washed 4 times with PBST (containing 0.1%Tween-20) at room temperature for 5 min each, dehydrated for 5 min with 25%, 50%, and 75% methanol (prepared by PBST), and then replaced twice with 100% methanol. Samples for immunofluorescence were fixed with 4% PFA and stored at 4 °C. Fluorescence in situ hybridization, and immunofluorescence samples were used for subsequent paraffin embedding and slicing.

Fish were fed daily with Fish Treasure fodder (Hayashikane Sangyo Co., Ltd., Ningbo, China). Secondary sand-filtered water was used in the experiments; pH was 8.0 ± 0.2, salinity was 23.5 ± 1, and water temperature was 22 ± 1 °C.

### 4.2. Morphological Observation of the “Pelvic Fin” of P. argenteus

The larvae and juveniles were anesthetized with MS-222 (Finquel MS-222, Sigma Inc., Marlborough, MA, USA) and sampled. Samples were collected daily during the larval period. The juveniles were sampled and observed every day until the fin rays were fully developed; thereafter, sampling was performed every 3–5 days, with more than 10 juveniles collected at each sampling. The morphological characteristics of each stage were observed under an Olympus stereomicroscope (Shibuya-ku, Japan), and the developmental sequence and characteristics of the fins were evaluated and recorded to clarify whether the lack of a “pelvic fin” in *P. argenteus* is a congenital or acquired degradation.

### 4.3. Scanning Electron Microscope Analysis

Samples used for SEM were rinsed 3–4 times with phosphate buffer saline (PBS), fixed in 3% glutaraldehyde solution for 12 h, and dehydrated for 10–15 min with 30%, 50%, 70%, 80%, and 90% gradient ethanol, and replaced twice with absolute ethanol. The tissues were then treated with ethyl alcohol and acetone in ratios of 3:1, 1:1, and 1:3, anhydrous acetone, and subjected to critical point drying using liquid carbon dioxide as the transitional fluid.

The larvae and juveniles of *P. argenteus* were fixed on the SEM stage with conductive adhesive, dried at 60 °C, and sprayed with gold using an ion sputtering instrument with spraying current of 10 mA and spraying twice for 50 s each time. The surface morphologies of the larvae and juveniles were observed using SEM (Hitachi S-3400, Tokyo, Japan).

### 4.4. Primer Designing

Two pairs of degenerate primers, Hoxd9-F1/ R1 and Hoxd9-F2/ R2, were designed to amplify the fragment of *Hoxd9* cDNA based on the core sequence of the *Hoxd9a* gene in the *P. argenteus* transcriptome database. The RACE primers, Hoxd9a-5′R1/R2 and Hoxd9a-3′F1/F2, were designed based on the core fragment to further amplify the full length of the *P. argenteus Hoxd9a* gene sequence. qPCR primers qHoxd9a-F1 and qHoxd9a-R1 were used for identification of *Hoxd9a* expression in different tissues and different periods. Based on the full length of *P. argenteus Hoxd9a*, specific fluorescent probe primers were designed for fluorescent in situ hybridization.

The primers used in the experiments are listed in Table 1. Except for the primers provided in the kit, the other primers were synthesized by Hangzhou Youkang Biological Co., Ltd (Hangzhou, China).

### 4.5. Extraction of Total RNA and Synthesis of First-Strand cDNA of P. argenteus

Total RNA was extracted from the fin rays of *P. argenteus* using an RNA Extraction Kit (Axygen). First-strand cDNA was synthesized using the PrimeScript RT Master Mix Kit (TaKaRa) with the total RNA of *P. argenteus* fins as a template. The synthesized cDNA product was stored at −20 °C or directly used for PCR.

### 4.6. Verification of the Target Fragment of the P. argenteus Hoxd9a Gene

The *Hoxd9a* cDNA core fragment was verified by PCR amplification using the specific primers, Hoxd9a-F1/ Hoxd9a-R1 and Hoxd9a-F2/ Hoxd9a-R2 (Table 1). The volume for PCR was 25 μL, and the optimized amplification conditions were as follows: initial denaturation at 95 °C for 3 min, followed by 35 cycles of denaturation at 94 °C for 30 s, annealing/extension at 60 °C for 30 s, and 72 °C for 1 min; the PCR products were stored at 4 °C. The PCR products were detected using 1.0% agarose gel electrophoresis. After the desired fragment was obtained, the target gene DNA fragment was purified using a DNA Recovery Kit (Axygen, Silicon Valley, CA, USA). After purification, the fragment was ligated to the pMD19-T (TaKaRa, Kyoto, Japan) vector, which was verified by transformation of competent cells (DH5α) to screen for positive bacteria, and then sent to Hangzhou Youkang Biological Company (Hangzhou, China) for sequencing.

### 4.7. Full-Length Sequence of the Hoxd9a Gene cDNA of P. argenteus

Sequencing results were obtained via the NCBI database (http://www.ncbi.nlm.nih.gov (accessed on 21 May 2022)) for BlastX analysis and comparisons with other species’ *Hoxd9a* sequence homology. According to the cloned core fragment of *Hoxd9a* gene, 3′RACE and 5′RACE-PCR primers were designed, including Hoxd9a-3′F1, Hoxd9a-3′F2, Hoxd9a-5′R1, and Hoxd9-5′R2, respectively.

The RACE template was synthesized according to the SMART^TM^ RACE cDNA kit (Clontech), and the 5′ and 3′ end sequences of the core fragment were amplified using RACE. After RACE-PCR amplification, the products were detected using 1.0% agarose gel electrophoresis, and sequencing verification was performed as described above. The core sequence, 5′RACE, and 3′-RACE sequences were compared and spliced with Vector NTI suite 11.5 software to obtain the full-length cDNA sequence of the *Hoxd9a* gene.

### 4.8. Real-Time PCR

Real-time PCR was performed to detect the expression of *Hoxd9a* in 8 distinct tissues and 12 different growth periods using cDNA tissues as templates and qHoxd9a-F1/R1 as primers. *18S rRNA* was employed as the internal reference gene in real-time PCR reactions with 2×UltraSYBR Mixture. The reaction protocol was 95 °C pre-denaturation for 10 min; 95 °C denaturation for 15 s; 60 °C annealing for 1 min, 72 °C extension for 25 s, 35 cycles. Graph Pad Prism9 statistical analysis software was used to examine the data, with three replicates for each sample and sterilized water as a blank control.

### 4.9. Hoxd9 Bioinformatics Analysis

The cloned *Hoxd9* sequence of *P. argenteus* was translated into amino acid sequences using ORF Finder (https://www.ncbi.nlm.nih.gov/orffinder/ (accessed on 21 May 2022)) online software, and on-line analysis of amino acid content, theoretical molecular weight, and isoelectric point of Hoxd9 protein by ProtParam (http://web.expasy.org/protparam/ (accessed on 21 May 2022)); Protscale (https://web.expasy.org/protscale/ (accessed on 21 May 2022)) was used to analyze the hydrophilicity and hydrophobicity of the Hoxd9 protein. PSORT (https://www.genscript.com/psort.html (accessed on 21 May 2022)) was used to analyze the nuclear localization of the Hoxd9 protein. Signal P-5.0 (http://www.cbs.dtu.dk/services/SignalP/ (accessed on 21 May 2022)) predicted the signal peptide and PSIPRED-4.02 software (http://bionf.cs.ucl.ac.uk/psipred/ (accessed on 21 May 2022)) was used to predict the secondary structure of the Hoxd9 protein. SWISS-MODEL (http://swissmodel.expasy.org/ (accessed on 21 May 2022)) predicted the tertiary structure of the Hoxd9 protein. Smart predicted of Hoxd9 domain (http://smart.embl-heidelberg.de/ (accessed on 21 May 2022)). The NCBI BLAST online tool was used to compare the homology of nucleic acid and amino acid sequences and search and download related sequences. The amino acid sequence of the Hoxd9 protein was analyzed using DNAMAN-5.2.9 software. The MEGA X software was used to compare the protein sequences and construct a phylogenetic tree using the neighbor-joining method.

### 4.10. Paraffin Embedding and Tissue Sectioning

Paraffin embedding was performed as follows. First, the wax injection flow rate of the embedding machine was adjusted to a moderate level, the embedding box was removed and placed on the hot table of the embedding table, and the first wax injection was opened. When embedding large specimens, the wax liquid level should not be higher than the upper edge of the mold groove; when embedding small specimens, an appropriate amount of wax should be placed on the embedding table. The tissue was placed into the mold in the correct manner, the embedding box was quickly covered after flattening, and wax was injected for the second time. The liquid level of the wax did not pass through the lower edge of the embedding box and was placed on a cold table. The embedded tissue wax block was cut continuously into 5 μm thick slices, and the slices were stored in a dry wooden box.

### 4.11. Fluorescence In Situ Hybridization Analysis of the Hoxd9a Gene in P. argenteus Larvae and Juveniles

In the non-conserved region of the cloned *Hoxd9a* cDNA sequence, a reverse probe was designed using Primer 5.0 (Table 1), and the specificity was detected via BLAST comparison. The fluorescent probe was synthesized using Wuhan Google Bio. *P. argenteus* slices in different periods were dewaxed, hydrated with gradient ethanol, and activated with 0.2% PBST for 30 min. The slices were incubated overnight with a fluorescent probe at 4 °C in the dark. The incubated slices were washed five times with DEPC-activated 1 × PBS for 10 min each. After washing, the slices were stained with DAPI at room temperature in the dark for 10 min. An anti-fluorescence quencher was added dropwise, and the slices were sealed and observed under a Nikon upright fluorescence microscope.

### 4.12. Immunofluorescence Localization Analysis of the Hoxd9 Protein in P. argenteus Larvae and Juveniles

According to the complete cDNA sequence and protein sequence of Hoxd9a, the DNA fragment of the *Hoxd9a* gene for prokaryotic expression was obtained using the whole gene synthesis method, which was cloned into the pEASY-Blunt vector. The positive clone was selected and verified, and then the recombinant plasmid was transformed into the *E. coli* Rosetta strain via the heat shock method, which could induce Hoxd9a recombinant protein by IPTG. New Zealand white rabbits were then immunized, and serum was purified to obtain high-quality antibodies. Hoxd9a antibody test chart of *P. argenteus* showed in Appendix A.

Immunofluorescence was performed as follow: *P. argenteus* slices were dewaxed at different periods and rinsed three times with PBS for 5 min each. After hydration with gradient ethanol, the slices were heated and boiled in repair solution (citric acid antigen) for 15 min, naturally cooled to room temperature, rinsed three times with PBS for 5 min each, and sealed with 5% BSA sealing solution at room temperature. The slice was removed from the solution, and excess liquid was removed using paper. Rabbit anti-PHB antibody (diluted with 3% BSA, 1:100) was added dropwise, and the slice was incubated overnight at 4 °C. The control group was incubated with blocking solution. After the slices were washed three times with 0.1% PBST for 15 min each, the sliced paper and liquid were removed. PHB (Alexa Fluor 488-Labeled Goat Anti-Rabbit IgG) diluted with 0.1% PBST 1:500 was added to the slice, which was incubated at room temperature for 1 h. Thereafter, the slice was washed six times with 0.1% PBST for 15 min each and removed from the light. The liquid was removed using paper. DAPI was added to the tissue slices for staining in the dark for 5 min. After rinsing with 0.1% PBST, an anti-fluorescence quencher was added dropwise to mount the slide, and a Nikon upright fluorescence microscope was used to observe and photograph the experimental results.

## 5. Conclusions

The full-length cDNA sequence of the *Hoxd9a* gene of *P. argenteus* was obtained using PCR and RACE. The open reading frame (ORF) of the gene was 633 bp and encoded 210 amino acids. Morphological analysis revealed no pelvic fin buds in the mesoderm of the lateral plate from the pectoral fin formation area to the tail of *P. argenteus* at each developmental stage. mRNA and protein localization analysis showed that Hoxd9a mRNA and protein were distributed in different developmental stages of *P. argenteus*; however, no fluorescence was observed in the abdomen, aligning with the morphological observations. In this study, the structure of the *Hoxd9a* gene, morphological characteristics at different developmental stages, and localization of *Hoxd9a* mRNA and Hoxd9 protein at different developmental stages of *P. argenteus* were assessed at the molecular level. Taken together, we found that the missing homeobox domain of Hoxd9a led to protein translation errors, which resulted in failed activation of the signaling pathway for pelvic fin formation initiation and development in *P. argenteus*.

## Figures and Tables

**Figure 1 ijms-24-11769-f001:**
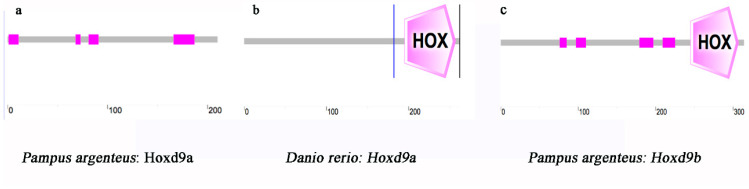
Prediction of the Hoxd9a/b domain of *P. argenteus* and comparison of the HOX domain (homeobox domain) with that of other fish. (**a**): prediction of the Hoxd9a domain of *P. argenteus*, pink boxes indicate low complexity domains, starting from left to right at 1–11, 78–73, 81–91, 166–187, respectively, but lack the homeobox domain; (**b**): prediction of the Hoxd9a of other fish represented by *Danio rerio*, where 195–297 is the homeobox domain; (**c**): prediction of the Hoxd9b domain of *P. argenteus*, where 245–307 is the homeobox domain and pink boxes indicate low complexity domains, starting from left to right at 76–85, 97–110, 179–197, 209–225, respectively.

**Figure 2 ijms-24-11769-f002:**
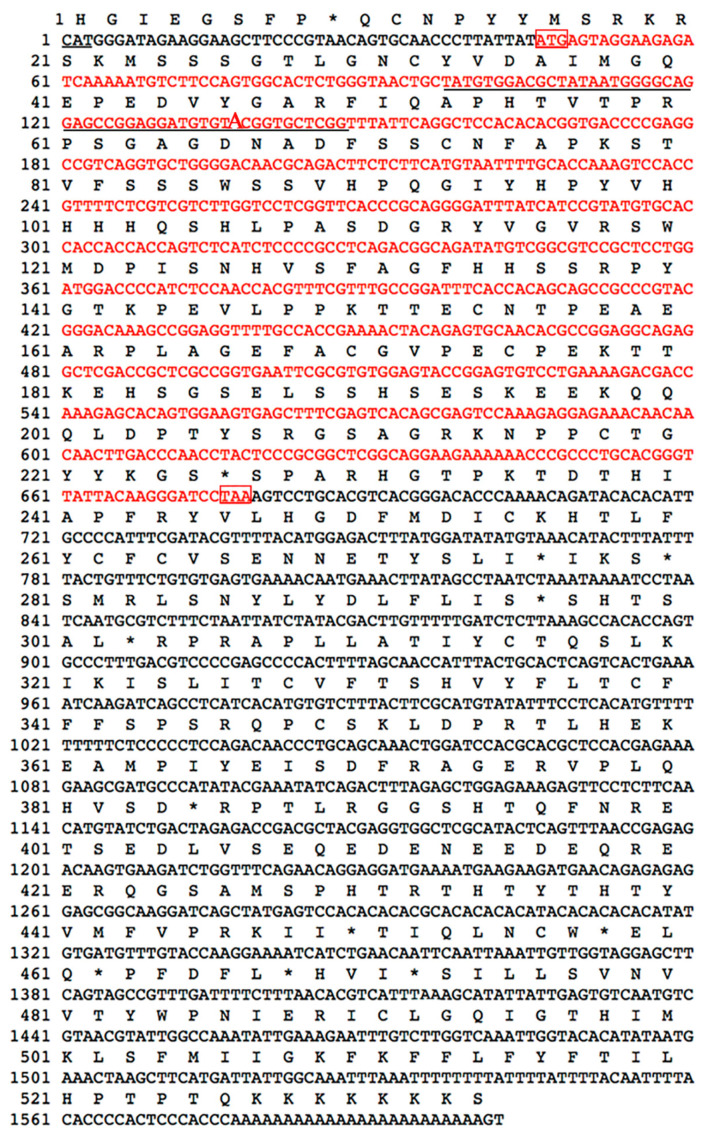
Full length and deduced amino acid sequence of *Hoxd9a* gene cDNA of *P. argenteus*. The red box sequence indicates the initiation codon and termination codon; Underline sequence indicates the promoter region (97–147 bp), and the transcription start shown in larger font; Red font indicates the *Hoxd9a* ORF; * indicates the end of protein translation.

**Figure 3 ijms-24-11769-f003:**
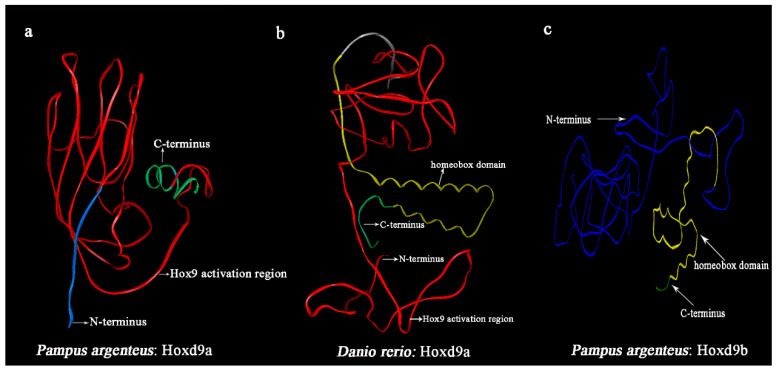
Prediction of tertiary structure of Hoxd9a protein in *P. argenteus* and comparison with tertiary structure of Hoxd9a protein of other fish. (**a**): The predicted tertiary structure of the *P. argenteus* Hoxd9a protein. The blue area is the N-terminus (1–11aa), the red area is the Hox9 activation region (12–189aa), and the green area is the C-terminus (190–210aa). *P. argenteus* Hoxd9a has no homeobox domain; (**b**): The predicted tertiary structure of Hoxd9a protein represented by *Danio rerio*. The red area is the Hox9 activation region (1–182aa), the yellow area is the homeobox domain (198–251aa), and the green area is the C-terminus (252–262aa); (**c**): The predicted tertiary structure of the *P. argenteus* Hoxd9b protein. The blue area is the N-terminus (1–244aa), the yellow area is the homeobox domain (245–307aa), and the green area is the C-terminus (308–314aa). *P. argenteus* Hoxd9b has no Hox9 activation region.

**Figure 4 ijms-24-11769-f004:**
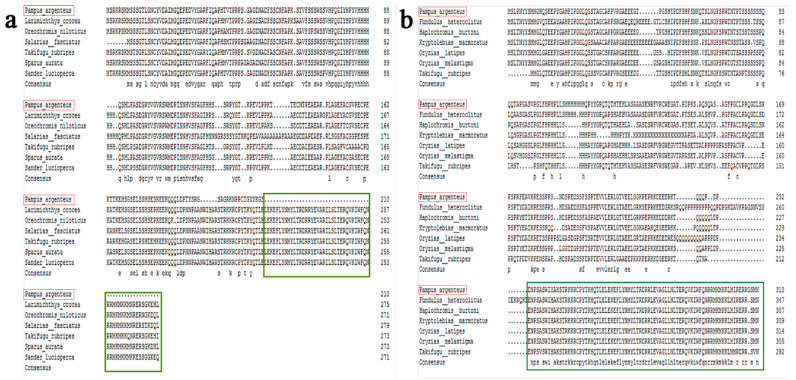
Multi-sequence alignment of the homologous sequences of *P. argenteus* Hoxd9 with those of six other fish species. The red box indicates the target species and the green box indicates the missing part of *P. argenteus* Hoxd9a, that is, the homeobox domain of 60 amino acid residues. (**a**): *P. argenteus* Hoxd9a; (**b**): *P. argenteus* Hoxd9b.

**Figure 5 ijms-24-11769-f005:**
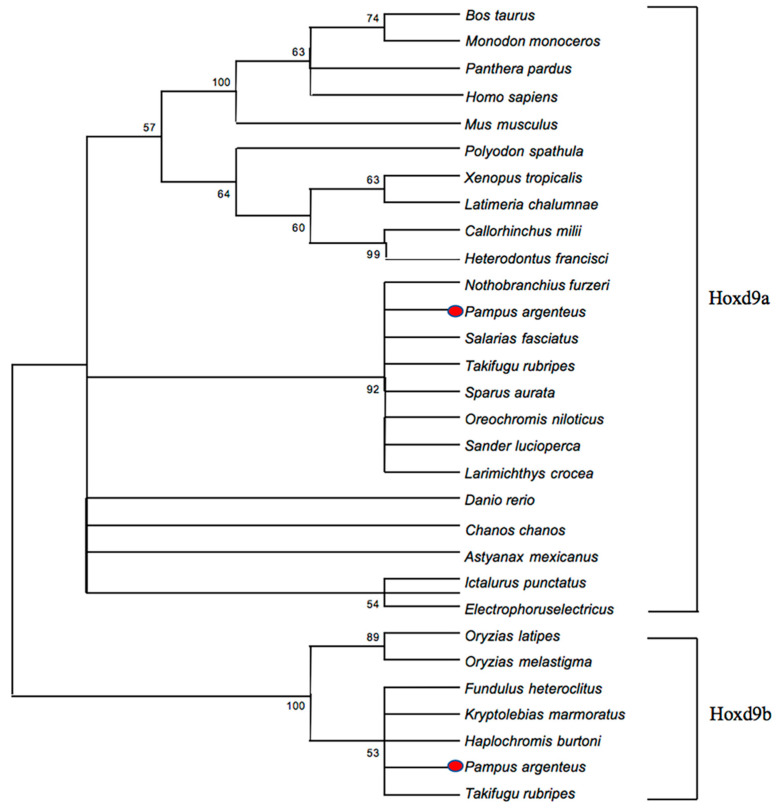
Hoxd9 phylogenetic tree of 28 species drawn according to the M-P method. Red dots indicate target species.

**Figure 6 ijms-24-11769-f006:**
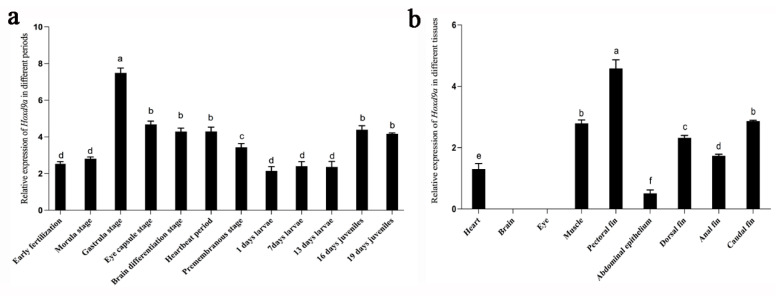
*Hoxd9a* mRNA expression in different tissues and growth stages of *Pampus argenteus.* Bars with different letters differ significantly (*p* < 0.05).

**Figure 7 ijms-24-11769-f007:**
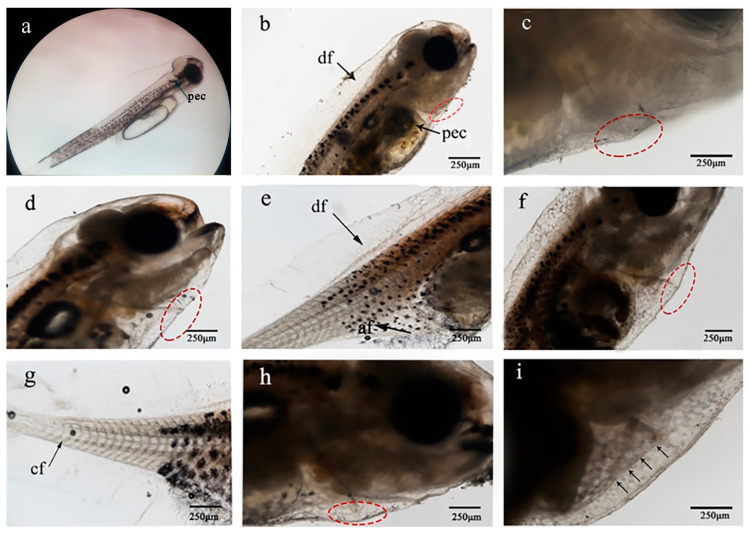
Morphological observations of *P. argenteus* “pelvic fin” development. (**a**–**i**): Optical micrographs of *P. argenteus* larvae. (**a**): 1 day larvae; (**b**,**c**): 7 days larvae; (**d**,**e**): 13 days larvae; (**f**,**g**): 16 days juveniles; (**h**,**i**): 19 days juveniles, the arrows indicate refined abdominal protrusion. The heads of all larvae are upright, and the abdomen of all larvae to the right. Scale bar = 250 μm. Abbreviations: af, anal fin; cf, caudal fin; df, dorsal fin; pec, pectoral fin; red dotted circle, abdominal protrusion.

**Figure 8 ijms-24-11769-f008:**
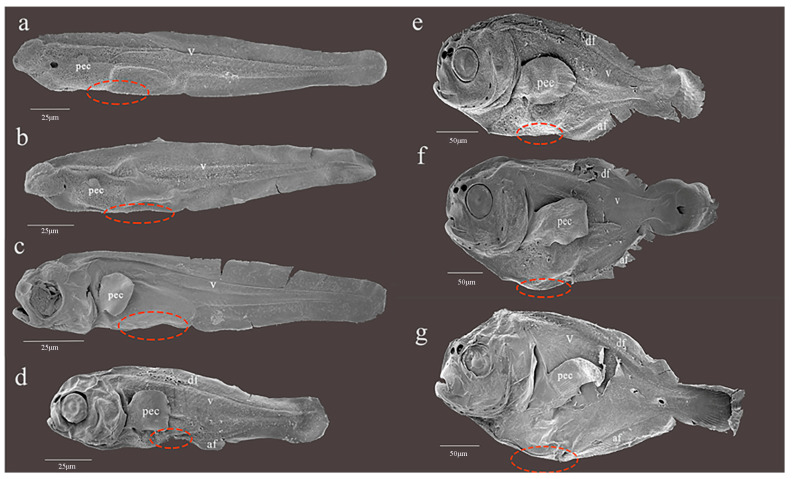
Scanning electron microscope observation of *P. argenteus* larvae. (**a**): 1 day larvae, SL 1.00 mm; (**b**): 3 days larvae, SL 1.00 mm; (**c**): 7 days larvae, SL 2.00 mm; (**d**): 13 days larvae, SL 3.00 mm; (**e**): 16 days juveniles, SL 5.00 mm; (**f**): 19 days juveniles, SL 5.00 mm; (**g**): 25 days juveniles, SL 5.00 mm. (**a**–**d**): Scale Bar = 25 μm; (**e**–**g**): Scale Bar = 50 μm. The red circle indicates the hypothetical pelvic fin formation area, abbreviated as v, vertebra; pec, pectoral fin; df, dorsal fin; af, anal fin.

**Figure 9 ijms-24-11769-f009:**
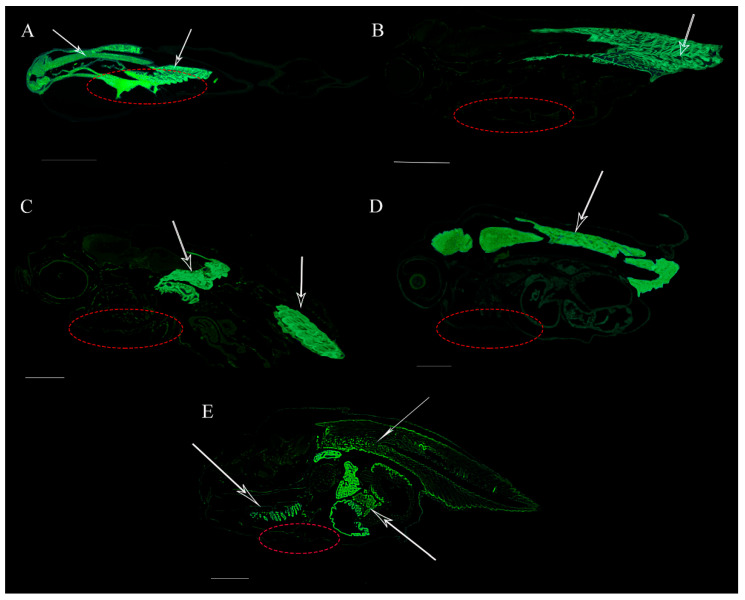
Fluorescence in situ hybridization of the *Hoxd9a* gene in *P. argenteus* at different developmental stages. (**A**): 1 day larvae, (**B**): 7 days larvae, (**C**): 13 days larvae, (**D**): 16 days juveniles, and (**E**): 19 days juveniles. Green fluorescence indicates the localization region of the *Hoxd9a* gene, indicated by an arrow. The red dotted circle indicates the hypothesized pelvic fin formation area. Bars = 500 μm.

**Figure 10 ijms-24-11769-f010:**
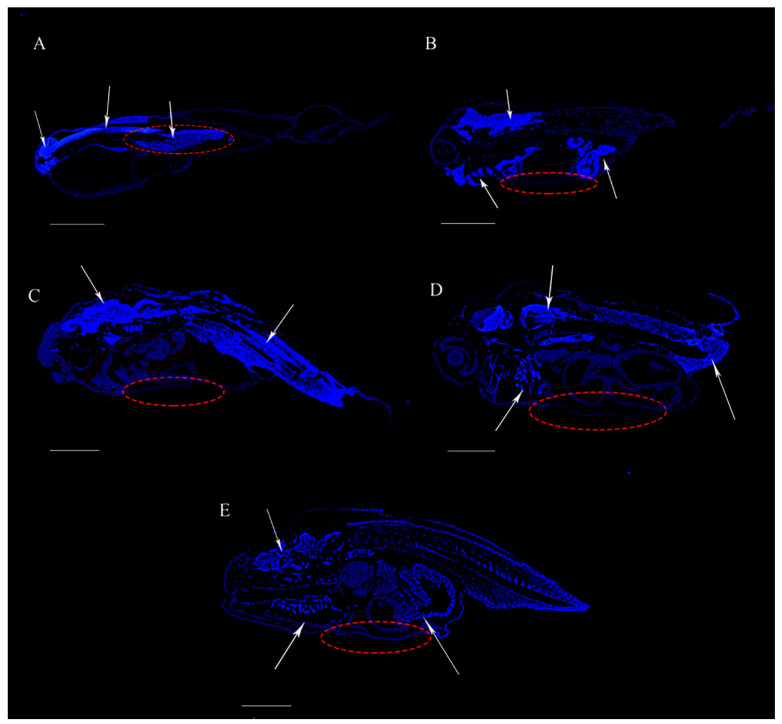
Immunofluorescence of the Hoxd9a protein in *P. argenteus* at different developmental stages. (**A**): 1 day larvae, (**B**): 7 days larvae, (**C**): 13 days larvae, (**D**): 16 days juveniles, and (**E**): 19 days juveniles. Blue fluorescence is the localization region of the Hoxd9a protein, indicated by arrow. The red dotted circle indicates the hypothesized pelvic fin formation area. Bars = 1000 μm.

**Table 1 ijms-24-11769-t001:** Oligonucleotide primers used in the experiments.

Primer Name	Sequence (5′-3′)
Degenerate primer	
Hoxd9a-F1	CTGGGTAACTGCTATGTGGACG
Hoxd9a-R1	GGTGGTGCACATACGGATGATA
Hoxd9a-F2	CCGAGCCCCACTTTTAGCAACC
Hoxd9a-R2	GGGAGTAGGTTGGGTCAAGT
RACE primer	
Hoxd9a-5′R1	CGTCCACATAGCAGTTACCC
Hoxd9a-5′R2	AAGAGAAGTCTGCGTTGTCCCC
Hoxd9a-3′F1	CAACTTGACCCAACCTACTCCC
Hoxd9a-3′F2	CTGCACGGGTTATTACAAGGGA
qPCR primer	
qHoxd9a-F1	TCTCGTCGTCTTGGTCCTCGGT
qHoxd9a-R1	GGTGAAATCCGGCAAACGAAAC
Fluorescent probe primer	
FISH-Hoxd9a	GAACAGAGAGAGGAGCGGCAAGG
Hoxd9a-control	CCTTGCCGCTCCTCTCTCTGTTC

## Data Availability

Not applicable.

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
