# Peer review of "Developmental Genetic Basis of *Hoxd9* Homeobox Domain Deletion in *Pampus argenteus* Pelvic Fin Deficiency"

_ijms, 2023, doi:10.3390/ijms241411769_

Round 1
Reviewer 1 Report
Dear Editors,
Dear Authors,
The manuscript entitled: “Developmental genetic basis of Hoxd9 homeobox domain deletion in Pampus argenteus pelvic fin deficiency” represents very interesting and valuable study that aimed to provide theoretical basis for the study of P. argenteus pelvic fin loss. The obtained results shed light on the evolutionary mechanisms of the pelvic appendages loss in the vertebrates. However, the reviewed study requires information clarification and corrections. Moderate editing of English language is required. All remarks about this have been placed in the attached file.
In conclusion, I recommend the manuscript for the publication after major revision. All remarks, questions and fixes were placed in the attached pdf file (yellow highlights contain fixes and sentence suggestions, while red highlights contain comments and questions).
Thank you for another interesting manuscript that I could review!

Dear Editors,
Dear Authors,
The manuscript entitled: “Developmental genetic basis of Hoxd9 homeobox domain deletion in Pampus argenteus pelvic fin deficiency” represents very interesting and valuable study that aimed to provide theoretical basis for the study of P. argenteus pelvic fin loss. The obtained results shed light on the evolutionary mechanisms of the pelvic appendages loss in the vertebrates. However, the reviewed study requires information clarification and corrections. Moderate editing of English language is required. All remarks about this have been placed in the attached file.
In conclusion, I recommend the manuscript for the publication after major revision. All remarks, questions and fixes were placed in the attached pdf file (yellow highlights contain fixes and sentence suggestions, while red highlights contain comments and questions).
Thank you for another interesting manuscript that I could review!
Reviewer 2 Report
The effects of Hox genes are known in many animal species, including fish. The study presented here describes the effects of Hox9 expression in a fish species.
While this is interesting from a general knowledge point of view, the implications of this discovery deserve more emphasis in the discussion.
In my opinion, the discussion needs to be reworked to discuss in greater depth the consequences of this discovery both from the point of view of general knowledge and of the consequences in this species, which is cultivated. The article would then gain in interest.
NA
Round 2
Reviewer 1 Report
Dear Editors,
Dear Authors,
The manuscript titled "Developmental Genetic Basis of Hoxd9 Homeobox Domain Deletion in Pampus argenteus Pelvic Fin Deficiency" has undergone significant improvements. The authors have diligently considered all of my remarks and suggestions. The manuscript now represents a high-quality study. I do not have any further suggestions, and I believe the manuscript is ready for publication in its current form. Congratulations to the authors for conducting such a meticulous and important study!